# A contrast sensitivity model of the human visual system in modern conditions for presenting video content

**Anastasia Mozhaeva**[1], **Michael J. Cree**[1], **Robert J. Durrant**[1], **Igor Vlasuyk**[2], **Aleksei Potashnikov**[2], **Vladimir Mazin**[2], **Lee Streeter**[1]*

**1** School of Engineering, The University of Waikato, Hamilton, New Zealand, **2** Department of Television and Sound Broadcasting, Moscow Technical University of Communications and Informatics, Moscow, Russian Federation

\* lee.streeter@waikato.ac.nz

**Data Availability Statement:** All relevant data are within the manuscript, Supporting Information files and open repository. The data underlying the

## Abstract

Digital video incurs many distortions during processing, compression, storage, and transmission, which can reduce perceived video quality. Developing adaptive video transmission methods that provide increased bandwidth and reduced storage space while preserving visual quality requires quality metrics that accurately describe how people perceive distortion. A severe problem for developing new video quality metrics is the limited data on how the early human visual system simultaneously processes spatial and temporal information. The problem is exacerbated by the fact that the few data collected in the middle of the last century do not consider current display equipment and are subject to medical intervention during collection, which does not guarantee a proper description of the conditions under which media content is currently consumed. In this paper, the 27840 thresholds of the visibility of spatio-temporal sinusoidal variations necessary to determine the artefacts that a human perceives were measured by a new method using different spatial sizes and temporal modulation rates. A multidimensional model of human contrast sensitivity in modern conditions of video content presentation is proposed based on new large-scale data obtained during the experiment. We demonstrate that the presented visibility model has a distinct advantage in predicting subjective video quality by testing with video quality metrics and including our and other visibility models against three publicly available video datasets.

## Introduction

Video streaming occupies a growing share of internet bandwidth, and in 2022 amounted to 82% of internet traffic. An Internet-enabled HD television drawing a couple of hours of content per day generates as much Internet traffic as the rest of the household today, on average. In 2023, 66% of connected flat-panel TV sets are 4K capable [1]. Therefore, compression and video transmission are important to reduce Internet transmission bandwidth, but video compression and transmission often are not visually lossless and lead to visible artefacts. The demands of modern video display systems go beyond the capability of transmission channels

results presented in the study are available from https://github.com/extezo/visibility-model.

**Funding:** The author(s) received no specific funding for this work.

**Competing interests:** The authors have declared that no competing interests exist.

to supply data to them. Therefore, adaptive video compression (methods without visually noticeable losses) is used today for modern ultra-high-resolution displays, virtual reality systems, etc. Video quality assessment is integral to adaptive video compression systems and adaptive image quality correction in media streaming.

To optimize video quality, one must first understand how human viewers perceive distortion caused by compression and transmission. Modern video compression is based on simple psychovisual models. Over the past decade, the focus of video quality research has shifted from the broad goal of understanding how people evaluate video quality to the more limited goal of developing computer algorithms that imitate the human subjective scores obtained in experiments [2]. In recent years, many methods have been introduced to assess video quality. However, an exceptionally high degree of prediction was achieved by a small number of video quality assessments (VQA) based on knowledge about aspects of the human visual system (HVS) that affect the perception of artefacts of the displayed signal [3–6].

VQA, based on knowledge of the characteristics of the human visual system, incorporates the results of psychophysical studies of human visual perception. However, the vast majority of research in visual psychophysics is principally aimed at gaining knowledge of how the human visual system operates: any relationship to video quality is usually secondary and typically not extensively discussed in such studies [2]. It is also common practice in psychophysical experiments (e.g. [7]) to standardize all but one variable of interest at a time, exacerbating the problem. The reductionistic laboratory-based approach minimizes and simplifies valuable analysis for understanding the features of the HVS in video perception but is unaware that by modifying multiple features associated with HVS understanding of a visual scene during data capture, one can, in principle, build a richer, and more realistic, model of HVS processing which, in turn, should enable the development of better automated VQA scoring that correlates well with subjective human scores.

All the above information and the complexity of existing experiments provide a paucity of available HVS data to describe how people perceive distortion on modern displays. In response, video quality researchers generate artificial datasets based on the limited available data, in which measurements have often been made using inconsistent protocols, detection criteria, viewing conditions and differences in stimuli [8]. Also, the currently available datasets derive from relatively early studies using fundamentally different display equipment. The results, therefore, do not guarantee a sufficiently accurate description of the conditions under which media content is now viewed. Consequently, designers of quality assessment algorithms and video compression must decide how psychophysical outcomes relate to quality in modern conditions (modern screens) for presenting video content without having a complete understanding of how presented video content is perceived by the human eye. Creating large-scale datasets of the human visual system acceptable for video quality designers remains an unresolved problem. New knowledge and tests on the relationship between the HVS and VQA are needed [2].

Previously, we proposed an experimental method for efficiently gathering data on the response of the low-level human visual system to video presented on modern displays. In the proposed paper, we offer the results of large-scale experiments of human visual perception using modern conditions for presenting video content. We measure the so-called threshold of visibility for temporal, spatial, and spatial-temporal variation at a large scale and derive a refinement of Watson's multidimensional visibility model [8] that is relevant to modern video display technology. We demonstrate an improved understanding of how HVS works through the simple collection of large amounts of new data. We demonstrate that new data improves the correlation of automated quality assessment with human subjective scores on modern

video by testing VQAs with the inclusion of our and other CSF models against three publicly available video datasets.

## Related work

The fundamental limits of the HVS were first described almost 60 years ago [9]. These limits were found through experimental and theoretical studies of the contrast threshold: the smallest contrast reliably perceivable under given conditions. Contrast sensitivity is defined as the inverse of the contrast threshold [8]. The contrast sensitivity function (CSF) is the modulation transfer function of the HVS and can be thought of as a bandpass filter [10]. The CSF characterises the earliest stage of vision that occurs 100 to 120 ms after the presentation of the stimulus to the observer's eye [11]. Nominally, early vision includes the capture, preprocessing, and coding of visual information and excludes interpretation or other cognitive processing of visual information [12]. (More recent definitions of early vision, for example, that given by Cecchi [11], include computation of basic properties like shape and color.)

The CSF is a bandpass filter that passes only visual stimuli that the observer can perceive. Only video artefacts in the passband region can be perceived by humans, hence the importance of CSF. Extant datasets on CSF have been collected with limitations. Datasets were collected firstly as a temporary measurement threshold [13]; secondly as the threshold for spatial contrast sensitivity only [14, 15]; and simultaneous spatial and temporal measurements using targets that were sinusoidal in both space and time [16]. Today, CSF research still does not present data sets with dependent spatial and temporal frequency measurements. The problem is that the visibility of temporal signals is probably not separable from their spatial configuration, so purely temporal measurements are of limited practical use [12]. Several efforts have been made to combine both spatial and temporal models of visibility [4, 17, 18].

Watson and Ahumada presented a linear model of space-temporal contrast sensitivity of the HVS, which they termed "the pyramid of visibility" [8]. The pyramid of visibility describes the spatial and temporal contrast sensitivity and its dependence on retinal illumination, all based on the modest results of much older studies [7, 13, 16, 19]. However, the model is limited in not covering the full spatial and temporal frequency range perceived by the human visual system. In 2021, the Computer Laboratory of the University of Cambridge, together with Facebook Reality Labs, presented the FovVideoVDP model [4], which is based on a combination of the spatial-chromatic CSF [20], the cortical magnification model and Kelly-Daly's model [21]. At the testing time, the video quality metric (VQM) with the FovVideoVDP model achieved the best correlation with human video quality assessment [22]. However, later was indicated that the FovVideoVDP model introduces a prediction error for low and high spatial frequencies. In 2022, a unified stelaCSF model was presented that considers all the main stimulus parameters, including spatial and temporal frequency and luminance. The model was generated by combining data from the early eleven measurements [22]. The use of older measurements of the CSF is due to the difficulty of conducting new large-scale experiments of the human visual system. To measure CSF in three dimensions for 10 points, in each measurement, where each measurement takes 90 seconds (which is typical for 30–50 trials), leads to $10 \times 3 \times 90/3600 = 25$ hours per experiment for one observer [22]. When using an optimized Bayesian adaptive method from medicine and psychophysics for studying thresholds of human perception of spatial and temporal change [23], it takes about 30 evaluations for a typical detection problem, or two hours to find 20 thresholds [24, 25]. However, suppose it is necessary to find ten mutual combinations between three parameters of brightness, spatial and temporal contrast thresholds. In that case, the Bayesian adaptive method requires 30,000 tests, amounting to 1 year of testing for one evaluator! The Bayesian adaptive method does not scale

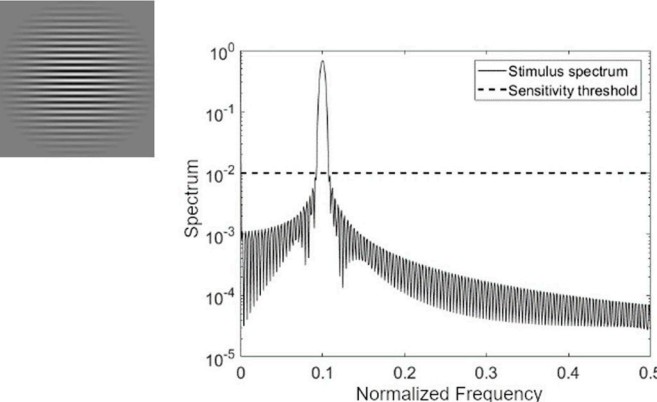

**Fig 1. The stimulus used to generate Mira.** The spectrum of the test signal, Normalized frequency, is the digital frequency of the spectrum. The horizontal dashed line represents the spatial sensitivity threshold [26].

up to fine-grained multi-feature data collection. Therefore creating a fairly comprehensive and sufficiently accurate model of the CSF faces the practical obstacle of large-scale and extremely time-consuming experiments. Using our recently proposed more efficient approach [26], we avoid the complexity of the implementation of the experiment and receive a large amount of data. Each participant requires about 30 minutes to perform 960 evaluations of various interactions of luminance, spatial, and temporal frequency values.

## Stimulus and apparatus

For our measurements, we employ a modern liquid crystal matrix display made using In-Plane-Switching (IPS) technology and computer control [26]. In our experiments, two tests were performed: the first using a uniform background with temporal flicker and the other a non-uniform test pattern with temporal flicker.

The non-uniform test pattern is the round sinusoidal lattice shown in Fig 1, dubbed "Mira". The test stimulus is round within the field of a clear vision so that all parameters of the test stimulus are symmetrical [26].

Let $w$ and $h$ be the pixel width and height of the display with centered coordinates $x_0$ and $y_0$ given by,

$$x_0 = x - \frac{w}{2}, \qquad \text{and} \qquad y_0 = y - \frac{h}{2}. \tag{1}$$

Let $d$ be the diameter of the test pattern in pixels, then $r$, the normalized centre-to-point distance, is given by

$$r = \frac{1}{d}\sqrt{x_0^2 + y_0^2}. \tag{2}$$

The Mira pixel brightness, normalized to the range [0, 1], is defined by

$$M(x_0, y_0) = \begin{cases} \frac{1}{2}\left[ \frac{I_0\left(\beta\sqrt{1-r^2}\right)}{I_0(\beta)} \sin\left(\frac{2\pi y_0}{T}\right) + 1 \right] & r(x_0, y_0) < 1 \\ \\ \frac{1}{2} & \text{otherwise} \end{cases} \tag{3}$$

where $\beta$ is a Kaiser window parameter, $T$ is the Mira period, and $I_0$ is the modified Bessel

function of the first kind. The Kaiser window function provides two improvements over the traditionally used Gabor stimulus. First, the Kaiser window has higher values for a wider region than the Gaussian function used in the Gabor function. The greater stimulus width ensures that more of the test participant's field of view is filled with the periodic pattern (this is important as we do not use a headrest or viewing aids in the experiments). Second (concordant with the first through Fourier theory), the Kaiser localizes more of the energy of the test signal in the main lobe, improving spectral specificity. In other words, the Kaiser window provides the largest stimulus with the smallest transition band [26] and, hence, optimised spatial frequency localization. The Kaiser window is multiplied by the sinusoidal pattern, providing a smooth transition from the pattern to the background region.

Note that in the experiments below, the display hardware limits the brightness values to 8-bit pixel values (0–black, 255–white) due to limited available equipment and can be changed on more advanced equipment. The pattern is presented in grayscale, and the pixel brightness value is set to a nominal level. The following sets of (8-bit) pixel brightness levels are used in our experiments: {40, 80, 120, 160, 200}. We use a linear grid of average screen brightness values, which is suitable for further analysis with existing visibility models [22]. The minimum and maximum average brightness values used ensure that the test pattern is not distorted by the hard limits of the eight-bit brightness value limits.

Now let $k$ be the spatial frequency, then the normalized spectrum $M(k_y)$ of the test object is given by,

$$M\left(k_y\right) = \frac{d \sinh\left(\frac{\pi}{\beta}\sqrt{\beta^2 - \pi^2 d^2 (k - \frac{2\pi}{T})^2}\right)}{2 I_0(\beta)\sqrt{\beta^2 - \pi^2 d^2 (k - \frac{2\pi}{T})^2}}, \tag{4}$$

Naturally, the limits of spatial resolution are determined by the test object's width and the screen display's pixel pitch.

The experiments in the proposed work use the limit method, in which the stimulus starts from zero (undetectable) modulation and is then gradually increased in intensity by the participant until the perceptual threshold of the stimulus is found. Finding the threshold is comparable to quality assessment metrics, where video content providers use solutions that only allow video of acceptable quality to the user [27]. Since we are measuring perceptual thresholds, there is little effect of the stimulus on the process.

The choice of liquid crystal technology enables the modulation of the backlight pixel brightness, hence control of temporal change. IPS matrices have well contrast a wide field of view [26]. We use an LG 22MP47A monitor (LG Corporation, Seoul, South Korea). The pixel brightness of the stimulus is sufficient to obtain representative results corresponding to the real conditions. The switching nature of LEDs makes linear dimming of the monitor backlight LEDs impossible. Therefore we introduce flicker via pulse-width-modulation (PWM) of the current flowing through the backlight LEDs (we also disabled the built-in PWM LED control of the monitor). The PWM temporal modulation control blinks the entire display at the specified rate (see Fig 2), [26]. PWM also provides near instantaneous response time of the flicker from the viewer's perspective (whereas each spatial pattern is static). The display's size and the participant's viewing distance allow us to assert that when the participant looks at the centre of Mira, the background region is not imaged in the zone of central vision.

In our experiments, the upper temporal frequency recorded by the eye was limited to 100 Hz. We get results in a limited range but avoid discomfort and potential hazards such as epileptic seizures [28]. We select a PWM bandwidth of 3.9 kHz, much higher than any temporal change detectable by the HVS, ensuring the accuracy of the flicker signal. The PWM

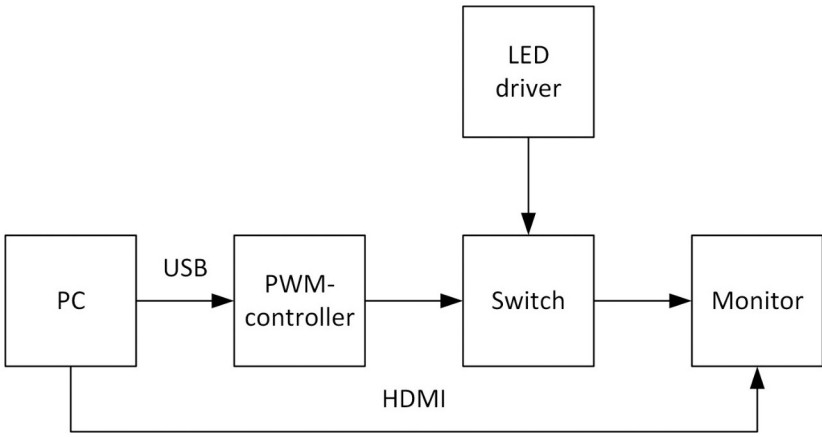

**Fig 2. Structural diagram of the display system installation for research [26].**

coefficient can represent the temporal component of the pixel brightness. Thus, visible pixel brightness is given by:

$$M(x, y, t) = m_{pwm}(t)M(x, y),\qquad(5)$$

where $m_{pwm}(t)$ is the PWM signal of temporal frequency $f$ Hz. We use a photodetector to confirm the correctness of the temporal signal as it emerged from the screen, see Fig 3, and an X-rite i1 Pro device to test the monitor color profile (d65). Using the presented equipment, we measured the relationship between the value of the specified pixel brightness and the screen's actual brightness used in the experiment. Fig 4 shows how it is possible to correspond the pixel brightness values used in the proposed work with the generally accepted brightness in candela per meter. The monitor was calibrated according to the recommendation of the International Telecommunications Union [29].

We remind the reader that these experiments are designed to quantify the perception of modern multimedia content, so we use the standard gamma value for video transmission systems of 2.4 (Rec. ITU-R Bt.601-4 standard). Displays are specified in terms of luminance, and consequently, knowing the form of the pyramid of visibility for a particular adapting luminance is needed when specifying rendering limits in space and time [8]. The screen is a 22-inch diagonal 1080p monitor with a 16:9 aspect ratio perpendicular to the view axis. The diameter of the stimulus is 0.2 m. Therefore, the maximum angular resolution of the human eye is one arc minute [30]. The period at the highest spatial frequency has 2 pixels, and the lowest is 540 pixels. In previous work [31], we calculated that the minimum allowable distance from the monitor for the stimulus to be in the clear vision zone is 0.872 m, and the maximum distance is 1.149 m [25].

## Method

Forty-two observers aged 20 to 40, with uncorrected vision, were recruited through the Moscow Technical University of Communications and Informatics. In the proposed work, normal vision is determined by typical user-generated content (in the Russian Federation, students at 16 years of age must undergo a general medical and physical examination, including eye testing). The participants do not use glasses, lenses, or other medical devices to correct their vision in normal daily activities, and the participants are free of known neurological disorders. Thirty-eight of the participants (90%) have no experience working with human perception of

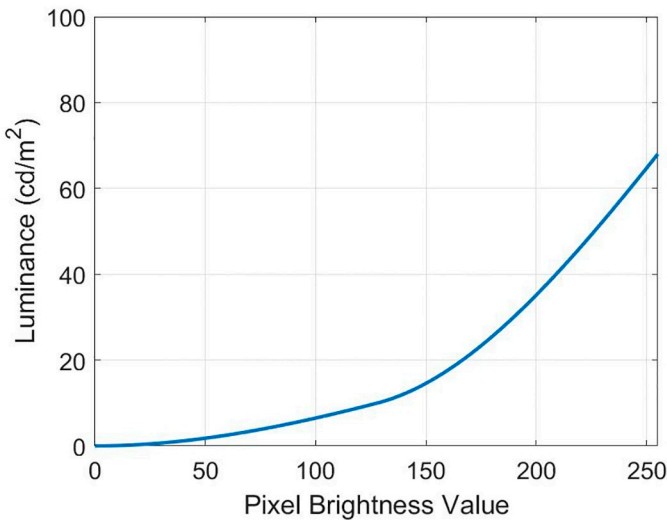

**Fig 3. Photocell with a linear light-signal characteristic connected to an oscilloscope.**

**Fig 4. Dependence of luminance in the immediate vicinity of the monitor on the pixel brightness value.**

visual information. The majority of participants were third-year undergraduate students, which is a good balance between three important parameters: physical maturity of the eye, daily use of typical user-generated content, namely video and image viewing on the internet, and lack of experience with work visual information perception. Of these three parameters, lack of experience with visual perception is especially important, as having such experience leads to improvement in artefact detection and spatial-temporal threshold detection [26]. The study was approved by the HECS Human Ethics Committee at the University of Waikato (HECS-20-64, where 21 December 2020 was the start, and 29 January 2021 was the end of the recruitment period for this study. Written informed consent was obtained from all participants).

The participant begins the test by pressing one of the buttons of a three-button mouse. The participant was provided a remote control with a rotatable controller and a button. The participant noted the needed test stimulus by rotating the control on the remote control and fixed the values by pressing the button on the remote control. The participant is not limited in time to view the test image, ensuring chromatic adaptation (the ability of the human visual system to adapt to changes in lighting in order to preserve the appearance of the colors of objects). As the tests were carried out at an individual pace, the next pattern was not presented until the participant recorded the answer by pressing a button on the remote control. The tests were presented to the participants in random order. We simulate, as close as possible, a normal environment for content consumption. Hence, these experiments represent a non-classical approach. Consequently, we do not use a head restraint or any viewing aids such as lenses. The same light was provided for all participants, the room had no external windows.

## First experiment: Temporal contrast sensitivity

A uniform grey screen (i.e. background only) with temporal flicker is presented to the participant, who then regulates the flicker amplitude to the minimum noticeable level, which is recorded. Contrast thresholds were measured at 12 different temporal frequencies and five different background levels. There were 13 participants (5 women and 8 men) who each recorded 60 levels. In total, 780 measurements were obtained (from all of the first experiment participants). The test took about 15 minutes per participant.

## Second experiment: Spatio-temporal contrast sensitivity

At first, the Mira pattern is not visible. The participants controlled the flicker amplitude, setting it to the minimum level that they could notice. Then, the participant adjusts the amplitude of the Mira pattern until they can minimally distinguish it from the background. Contrast thresholds were measured at eight different spatial and twelve temporal frequencies at five different pixel brightness levels. Each of the 29 participants (5 women and 24 men) performed 960 threshold evaluations. A total of 27840 estimates were acquired from the second experiment. The testing for the second experiment was about 30 minutes per participant.

The trials were undertaken at the participants own pace in an effort to reduce any side effects of fatigue, and the participant was allowed to take a rest break at any time. A 95% confidence interval for our data was used. The standard deviation $\sigma_{kfl}$ to evaluate the confidence interval for each presentation is given in the Rec. ITU-T Bt.500-15 standard [32]. All experiments continued until the confidence interval fell below 5% of the current mean value for each point across participants. Consequently, the basis of the sample size for the 1st and 2nd experiments was decided according to Rec. ITU-T Bt.500-15 standard [32]. In other words, the experiment for a fixed ($f$, $l$, $k$) is repeated until the validity criterion is satisfied. Each triplet, i.e. the value of $s$ at each ($f$, $l$, $k$), was tested and halted independently of every other triplet, where,

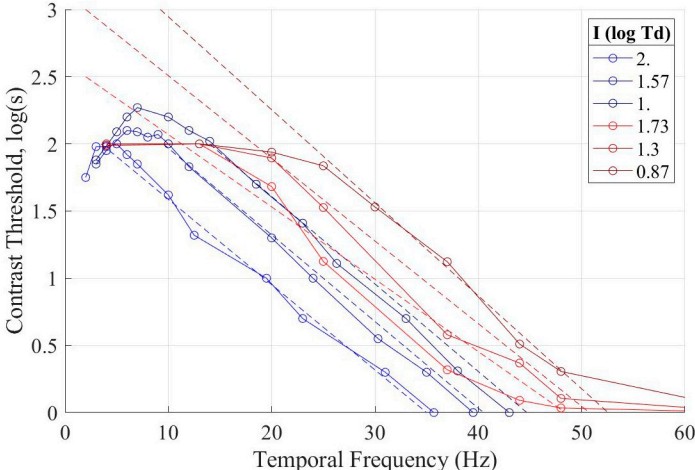

**Fig 5. The results for the temporal components of responses of the participants in Experiment 1 compared to the linear model and pyramid of visibility, dashed lines are the linear model while solid lines are from data points, where red data from Experiment 1 and blue from Watson's work.**

as above, *k*, is spatial frequency; *l* is background pixel brightness; and *f* is the temporal frequency of the flicker.

## Results

### Temporal contrast sensitivity

In Fig 5 we present the temporal (flicker) frequency results from Experiment 1. We see the linear fit of the contrast threshold with temporal frequency. When compared with Watson's work, namely the graphs with de Lange's results [8], we see that the model of temporal contrast sensitivity has a similar shape as previous research where modern conditions were not considered. Watson gets a slope of -0.064 for the log contrast sensitivity against temporal frequency. In the presented case, the slope is -0.058. However, the fit is only good with a narrow range, where the range of good fit depends on illuminance.

### Spatio-temporal contrast sensitivity

Figs 6 and 7 show graphs of the spatial-temporal contrast sensitivity of human visual perception (Experiment 2). The contrast threshold is near-linearly proportionate to spatial frequency but non-linear with temporal frequency. Spatial-frequency characteristics of the visual system, as shown earlier by Kulikowski and Watson's work, are well approximated by a linear function for spatial frequencies above 5-7 cycles/degree [7, 8], with minimal change from one temporal frequency to another. In contrast, representing the temporal characteristics of the visual system using linear functions under the conditions of the experiment is only possible within a limited interval of values of the temporal frequency. This nonlinearity is in contrast to the earlier work of Kulikowski, who intervened with feedback to stabilize the observed visual system parameters [7]; and Robson, who used binoculars [33]. We employed neither measure. It can be assumed that such a difference in comparison with previous works is because the earlier experiments had independent goals of separately detecting the spatial and temporal threshold. Moreover, in Watson's work, each of the spatial and temporal components was fixed, and the

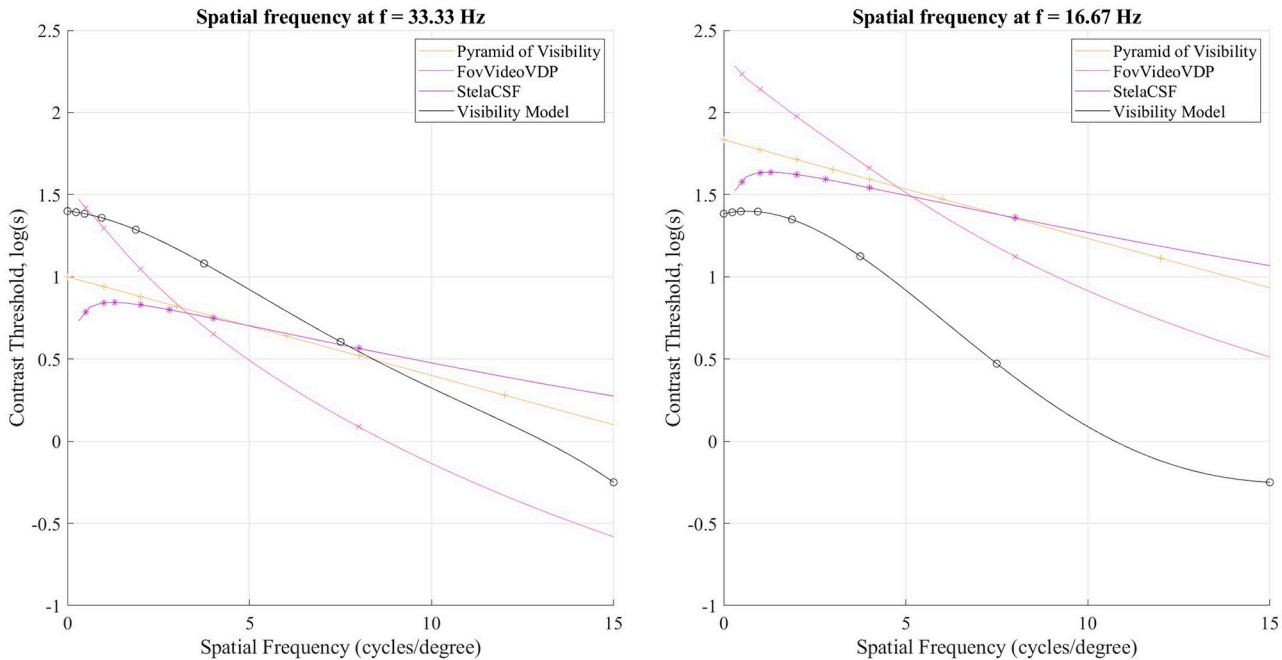

**Fig 6. Contrast sensitivity as a function of spatial frequency at several temporal frequencies from Experiment 2 compared to the pyramid of visibility [8], FovVideoVDP [4], stelaCSF [22].**

other varied [8]. We employed viewing conditions more similar to real-world media consumption.

It should also be noted that participants noticed flickering at the higher frequencies tested at 15, 40 Hz (pixel brightness level is 200), at 36 Hz (pixel brightness level is 160), 13, 20 and 48 Hz (pixel brightness level is 80); 40 and 46 Hz (pixel brightness level is 40). These values were reasonably excluded from the analysis; details are discussed below.

## The model of visibility

We present our CSF model (Visibility model) of the first part of early vision, namely that of the filtering stage, which governs what spatio-temporal fluctuations in stimuli the HVS responds to. Fig 8 shows average real values for all scores of Experiment 2 at a pixel brightness stimulus level of 120. From the previous section, a linear pyramid of visibility model may be appropriate for spatial frequency dependence but not for temporal frequency. Therefore, we develop a polynomial approximation of the model based on the results from the previous sections. The model is found by linear regression against polynomial terms. We find by inspection that a 4th-order polynomial strikes an appropriate balance between encompassing non-linearity and minimising over-fitting:

$$\log(s)(k', f', l') = \sum_{\alpha, \gamma, \delta: \alpha + \gamma + \delta \leq 4} c_{\alpha, \gamma, \delta} k'^{\alpha} f'^{\gamma} l'^{\delta}. \tag{6}$$

where $K'$, $f'$, $l'$ are normalized parameters setting the range of each from zero to one (in our experiments), i.e. the maximum spatial frequency, in our experiments, is 15 per deg, the maximum temporal frequency is 66.6 Hz, and the maximum background pixel brightness is 200.

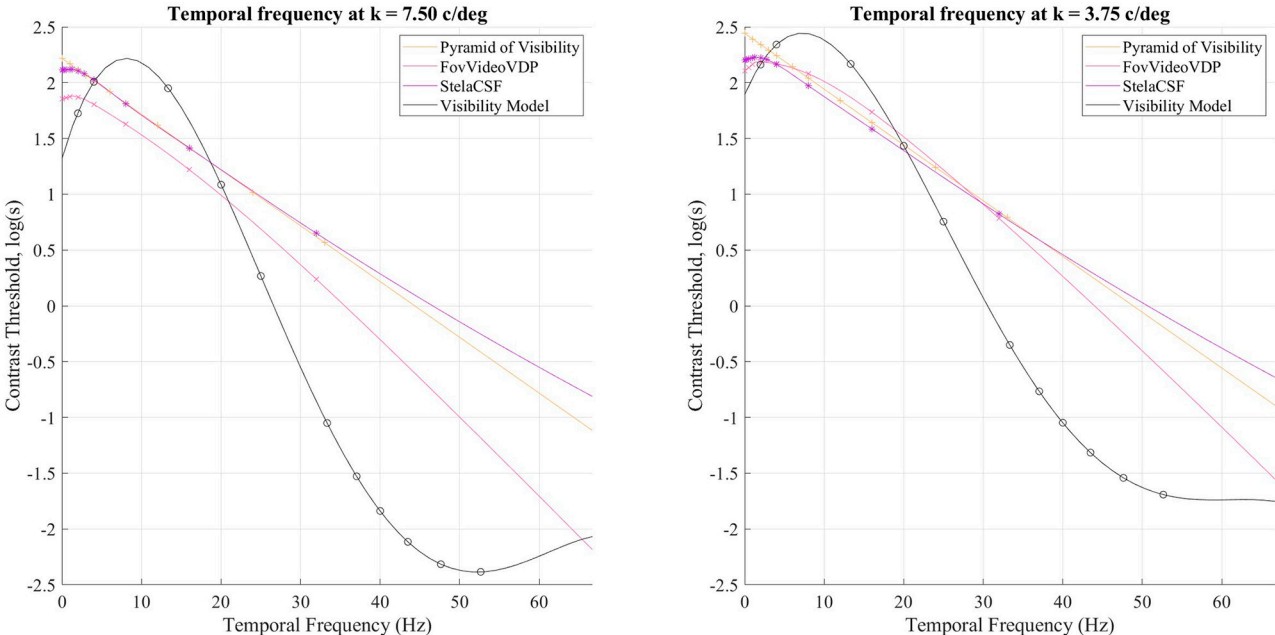

**Fig 7. Contrast sensitivity as a function of temporal frequency at several spatial frequencies in Experiment 2 compared the pyramid of visibility [8], FovVideoVDP [4], stelaCSF [22].**

Eq 6 may be written in the following matrix form

$$(\log(s)(k',f',l') = Xc^T, \tag{7}$$

where, now, $\log(s)$ is a vector of the logarithm of the measured $s$ values, $X$ comprises the polynomial terms, and $c$ the vector where the correspondence of the terms. In practice, we solve for $c$ via. ridge regression to mitigate overfitting. We report the coefficients of the defined approximation polynomials on our open repository., viz.:

$$c = (X_i^T X_i + \lambda I')^{-1} X_i^T y, \tag{8}$$

where $\lambda = 0.001$ is a regularization coefficient, $I'$ is a modified identity matrix except that $I'(0,0) = 0$, $y$ is the vector of the logarithm of the average responses of the participants, and each row of the $X_i$ matrix is the $X$ matrix with combinations of $l, f, k$ for the $i$th experiment.

## Discussion

Comparison of existing CSF models and the model presented in this paper on existing older CSF datasets by best-fit estimation is impossible. The data on the temporal aspect in modern conditions have critical differences from the older ones.

At this stage of technology development, two models of CSF generated from older research exist and consider all the basic aspects of human perception that can improve the performance of video quality assessments. The contrast sensitivity model used in the FovVideoVDP VQM [4], described above, and stelaCSF, introduced in 2022. Data from 11 early studies were pooled to modelling stelaCSF [22]. The stelaCSF approach can predict data from all open early studies using the same set of parameters and has accurate fit predictions over the entire range, including low frequencies. The results show that stelaCSF can explain datasets better than existing models, regardless of the number of dimensions considered, including the pyramid of visibility

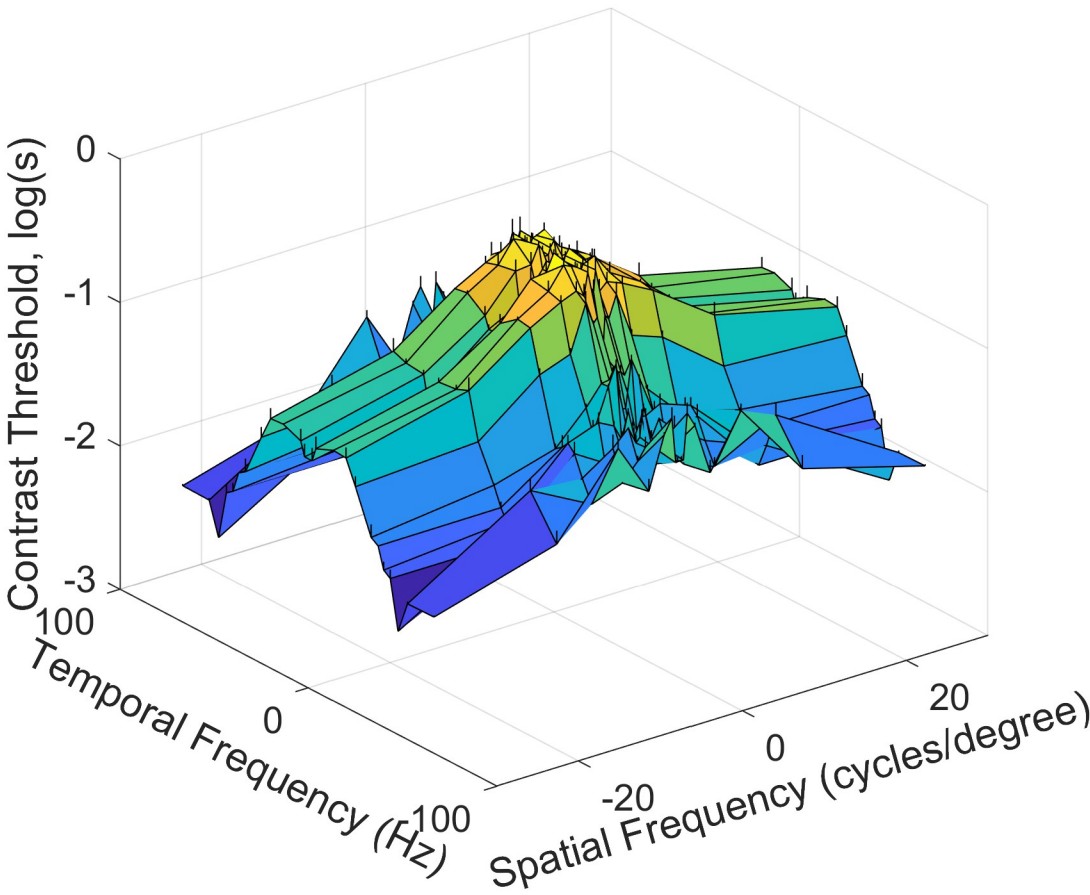

**Fig 8. The average of measured values for all evaluations log(*s*) of Experiment 2 (120 pixel brightness level).**

[22]. We remind the reader that the model presented in the paper is a refined visibility pyramid for evaluating video content in modern conditions, Fig 9.

This section analyses two VQMs [3, 4], with different visibility models included in both metrics. In the first VQM, the inclusion of CSF models is based on the calculation of the weighted PSNR, Fig 10. All three models generate time coefficients that change with the frame, which are then passed to the VQM to calculate the score value. We averaged quality predictions across all frames in each video sequence in the data sets. We fit a non-linear model to

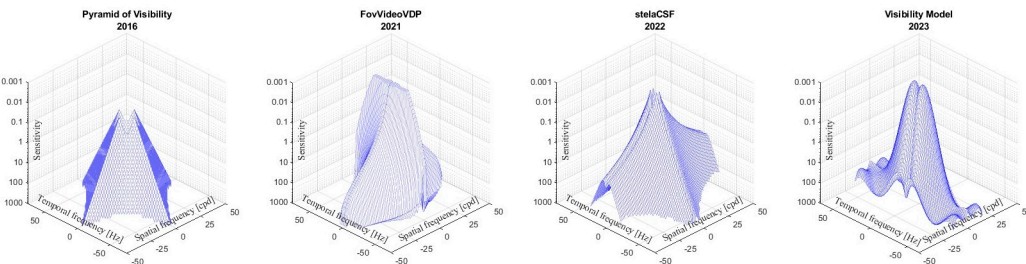

**Fig 9.** Visualization of contrast sensitivity models (from left to right): Pyramid of visibility [8], FovVideoVDP [4], stelaCSF [22], Visibility model.

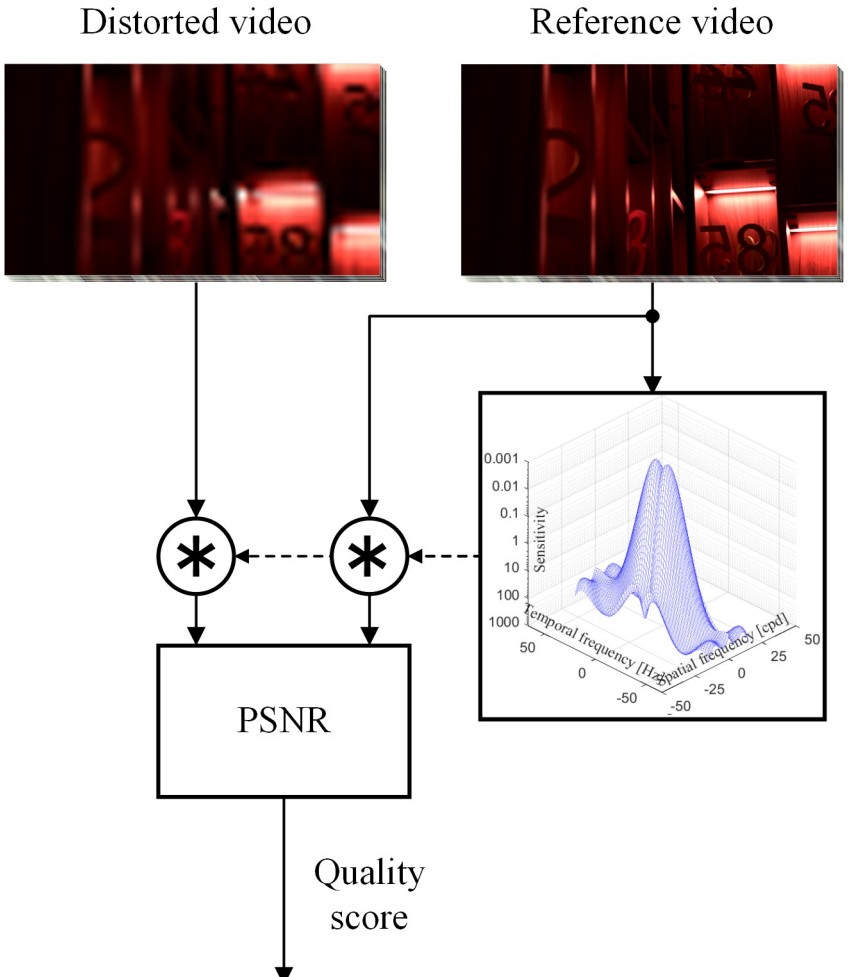

**Fig 10. Block diagram PSNR inclusion of stelaCSF model [22], where ∗ is the convolution.**

each metric that maps from predictions of the average human scores in the open datasets. We report the prediction results as the Pearson correlation coefficient (PLCC). We also report the average of PLCCs computed per dataset, including upper and lower bounds.

All models' background brightness was set at 120 cd/m². The resulting contrast values for the stelaCSF and FovVideoVDP models were converted from Weber to logarithmic. The parameters of the stelaCSF model and the FovVideoVDP model, except for those previously described, such as orientation and eccentricity, were set to "0". The overall performance for all models is shown in Table 1.

The following databases were used to test visibility models: LIVE Netflix QoE database consists of 112 distorted videos created from 14 footage at 1080p at 24, 25 and 30 fps by overlaying a set of 8 different playback templates [34, 35]; MCL-V contains 12 original video clips and 96 distorted video clips with subjective evaluation scores [36]; CSQ database is a large-scale set of encoded videos with constant subjective assessment. The database uses H.264 compression, and the number of artefacts is 44800 [31].

For the LIVE Netflix QoE database, the correlation value is calculated between frame-by-frame metric values and frame-by-frame subjective evaluation values. In the MCL-V database,

**Table 1. Comparison of the video quality metrics.** Pearson correlation coefficient (PLCC).

| Pearson Linear Correlation Coefficient for VQA/Database | LIVE-NFLX | | | MCL-V | | | CSQ | | |
|---|---|---|---|---|---|---|---|---|---|
| | Upper | PLCC | Lower | Upper | PLCC | Lower | Upper | PLCC | Lower |
| PSNR | 0.34 | 0.318 | 0.297 | 0.307 | 0.110 | -0.292 | 0.453 | 0.388 | 0.317 |
| PSNR-M with stelaCSF | 0.384 | 0.364 | 0.352 | 0.628 | 0.481 | 0.302 | 0.42 | 0.404 | 0.393 |
| PSNR-M with FovVideoVDP | 0.402 | 0.383 | 0.368 | 0.64 | 0.497 | 0.321 | 0.381 | 0.363 | 0.343 |
| PSNR-M with Visibility model | 0.422 | 0.401 | 0.381 | 0.643 | 0.502 | 0.36 | 0.458 | 0.437 | 0.426 |
| FovVideoVDP with stelaCSF | 0.448 | 0.429 | 0.426 | 0.339 | 0.174 | -0.038 | 0.397 | 0.381 | 0.375 |
| FovVideoVDP with FovVideoVDP | 0.342 | 0.318 | 0.292 | 0.304 | 0.100 | -0.113 | 0.397 | 0.382 | 0.376 |
| FovVideoVDP with Visibility model | 0.49 | 0.466 | 0.44 | 0.29 | 0.121 | -0.054 | 0.34 | 0.339 | 0.328 |

the average value per video represents the subjective rating. We find the average value of the VQM for each video, after which we calculate the correlation between the average values of the VQM and the average values of the subjective rating. The content in the CSQ database is a series of videos joined together to form a consistent quality sequence. For each of these videos, the average value of the VQM and the average value of the subjective opinion is found, after which the correlation between the average values.

Of the models tested, the VQM with the new unique model was found to have the best and comparable results in predicting subjective video quality. The presented model has a clear advantage in terms of the scale of the data required at this stage of technology development to create VQMs based on machine learning.

In the proposed work, the comparison is not made for all parameters in the stelaCSF model. This is due to the peculiarity of the comparison since the main goal of the proposed work is to demonstrate the advantages of the obtained data on temporal contrast sensitivity in modern conditions for the provision of media content. The presented model considers only the area of central vision, and we need to develop a similar spatio-temporal model of peripheral vision in future work.

It should also be noted that the viewing conditions in older studies significantly differ from the displays in common use today. First, the field of view was very small, which is not characteristic of modern video presentations. Second, the Kinescope, a display based on cathode-ray tube (CRT) technology, was used in older studies. The main disadvantages of CRTs are well known: low static contrast, low spatial resolution, dependence of the point scattering function on brightness, low stability of CRT parameters over time (due to changes in cathode emission and phosphor luminosity, uneven brightness and contrast across the field) [37]. It should be noted that, to a certain extent, the shortcomings are compensated in more real precision display devices, but usually to obtain the best user experience rather than improve parameters and characteristics. We consider the most significant disadvantage of CRT stimulus formation to be the pulsed nature of image formation, where the electron beam scans the screen, and its intensity is modulated, thus transmitting the brightness of each point to the screen. The decrease in luminescence intensity concerning the afterglow time (the phosphor continues to glow for some time after the beam has left) is considered exponential. Therefore, to avoid temporal distortion of the stimulus due to a long afterglow, it is necessary to use a phosphor with a short afterglow time. The temporal stimulus is a short pulse of light with a longer but also short decay shift in time across the CRT screen. In assessing the sensation of the brightness of such a stimulus, possible nonlinearities in the perception and the effects associated with the afterglow duration are usually not considered. For example, the prolonged perception of an image on a CRT monitor causes observer fatigue more than a similar static image illuminated

by a constant illumination source over time. Estimating the degree of measurement error in stimulus synthesis by the method used when using a different display-based is beyond the scope of this work.

The advantage of a CRT is the absence of a pixel structure, the formation of a stimulus not by a triad of primary colours but by a phosphor colour predetermined during production. However, as mentioned above, the most significant disadvantage of the CRT-based setup used in the past is its relatively long time of subjective experiment. Today, with the development of statistical apparatus and data science to create accurate models, it is desirable to have a large amount of initial data for processing since the model's simplicity and its design are no longer significant factors determining its popularity in practical use. At the same time, recreating a CRT-based installation seems irrational, given the limitations of such a display device listed above. Also, such devices have long been discontinued, and experiments need to use a custom-made CRT with high spatial resolution and, at the same time, a high sweep rate. Based on a balanced decision about the shortcomings and advantages of today, it was decided to manufacture a display device based on a serial display.

In Fig 11, we display the system model for different pixel brightness stimulus levels which shows that, while the overall contrast threshold changes with pixel brightness, the shape of the contrast threshold characteristic depends little on pixel brightness. It is apparent that, as temporal frequency increases, the spatial frequencies that are easier to see become less noticeable more quickly than those spatial frequencies that are more difficult to distinguish from the background. At less noticeable spatial frequencies, the temporal frequency is the dominant factor in terms of decline in the contrast threshold, in agreement with much earlier work [28]. During the experiment, we considered increasing the number of brightness points, but the degree of nonlinearity was not adequately large enough to require more.

The multivariate model presented herein describes the interaction between the spatial and temporal frequency response of the HVS over a broad range of relevant values. While recent

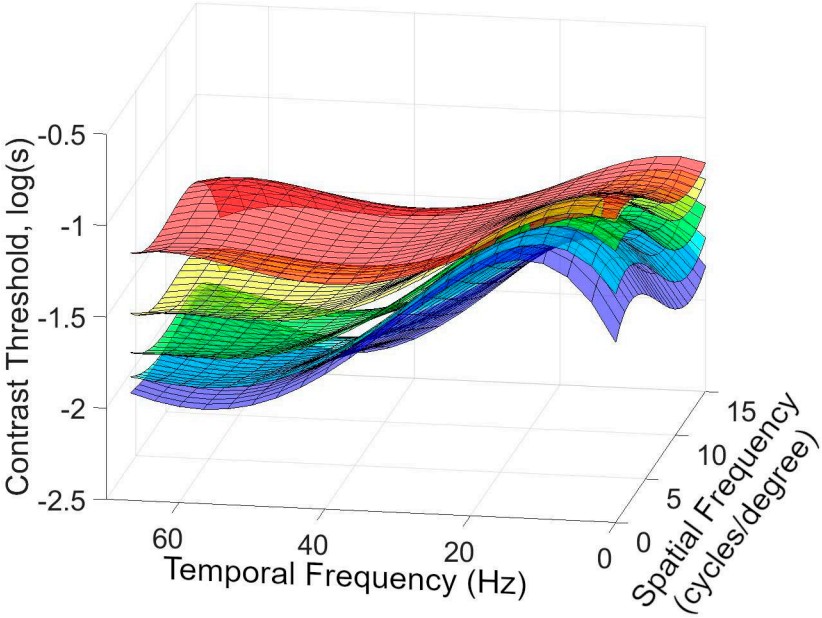

**Fig 11. The model of visibility is built from the values of Experiment 2 with a pixel brightness stimulus level of red-40, yellow-80, green-120, blue-160, purple- 200 where s = 0.1.**

evidence exists that the HVS can perceive temporal changes up to 500 Hz under controlled conditions [38], our results show that under conditions more closely resembling real-world content consumption, there is little response beyond 200 Hz flicker. The bandwidth of our equipment is 3.9 kHz, which is faster than humans perceive flicker artefacts [38]. Therefore, the principal source of measurement error is (as elsewhere) imperfection in the test apparatus and the measurement process.

The control features and elements of an IPS matrix monitor, in combination with PWM modulation of the backlight pixel brightness, can create spurious maxima of the spatio-temporal spectrum of the displayed stimulus. The results show that participants noticed flickering at the higher frequencies tested. In other words, they saw these spurious maxima, causing unexpected changes in the threshold values. Based on the exact repeatability of the frequencies of these anomalies, we assumed that the recorded effect is associated more with the test equipment's imperfections than with the visual system's properties. We did additional measurements using a photosensor and an oscilloscope, but no spurious frequencies were detected in the generated stimulus. Also, it must be considered that at these frequencies, the contrast sensitivity of vision becomes commensurate with the value of the quantization noise of the display device and the PWM backlight controller, as well as with the sampling parameters of the measuring equipment. We excluded these changes when processing the results and provided the experimental data for further research. (However, we also note that excluding these threshold values caused very little change in our CSF model.) Prolonged display flickering at high amplitude may trigger seizures in people not diagnosed with photosensitive epilepsy. Hence, for participant safety, we did not perform reverse threshold measurements with a transition from high modulation to zero. (Clinical practice shows that 76 percent of photosensitive people do not know about their photosensitivity [39].) Also, as our goal was to bring the experiments as close as possible to the typical content consumption, a head holder was not used. Regardless, participants were asked to keep their heads within a fairly narrow range of positions. Consequently, any head movement is compensated for by large-scale experiments and testing of significance: 960 thresholds are found in the experiment for one participant, terminating when significance is obtained.

Limitations of the experiment include imperfections in the test apparatus and the measurement process. Due to the limitation of contrast and maximum screen pixel brightness, the characteristics of vision in the state of its full adaptation were studied. Changes in the mode of a non-adapted HVS, which can be used, for example, when observing high-contrast scenes in HDR systems, require an appropriate stimulus generation device.

## Conclusion

In this work, new large-scale experiments are proposed on visual perception and measurement of the characteristics of the human visual system using a custom testing system. In total, 38220 measurements were taken of the visibility thresholds as temporal and spatial-temporal parameters changed. The multidimensional CSF model of the human visual system in modern conditions for presenting video content is presented. It was shown that for high spatial frequency, there is linear proportionality between the logarithmic contrast threshold and the spatial frequency for modern multimedia devices but nonlinear for temporal frequencies. We demonstrated the need for data collection in a modern video-viewing environment by comparing our new model with current models. Our new model will see the application in modern visualization technologies. We believe the new knowledge can significantly advance progress in video compression and quality, visualisation systems, and video masking. Future work will continue to see the integration of our CSF model into non-reference video quality metrics to improve

bandwidth and reduce storage space. We also need to develop a similar spatio-temporal model of peripheral vision, for which the rapidly developing field of virtual reality devices provides a promising experimental pathway.

## Acknowledgments

The authors thank John A. Perrone and Vincent Reid for helpful discussion and reading of this work.

## Author Contributions

**Conceptualization:** Anastasia Mozhaeva, Lee Streeter.

**Data curation:** Anastasia Mozhaeva, Michael J. Cree, Lee Streeter.

**Formal analysis:** Anastasia Mozhaeva, Michael J. Cree, Robert J. Durrant, Lee Streeter.

**Methodology:** Anastasia Mozhaeva, Igor Vlasuyk, Lee Streeter.

**Project administration:** Anastasia Mozhaeva, Lee Streeter.

**Software:** Anastasia Mozhaeva, Aleksei Potashnikov, Vladimir Mazin.

**Supervision:** Michael J. Cree, Robert J. Durrant, Igor Vlasuyk, Lee Streeter.

**Validation:** Anastasia Mozhaeva, Michael J. Cree, Robert J. Durrant, Igor Vlasuyk, Lee Streeter.

**Visualization:** Anastasia Mozhaeva, Vladimir Mazin.

**Writing – original draft:** Anastasia Mozhaeva.

**Writing – review & editing:** Michael J. Cree, Robert J. Durrant, Lee Streeter.

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
