## [Decision Letter · Decision Letter 0]

12 Feb 2024

PONE-D-23-30415A contrast sensitivity model of the human visual system in modern conditions for presenting video content.PLOS ONE

Dear Dr. Mozhaeva,

Thank you for submitting your manuscript to PLOS ONE. After careful consideration, we feel that it has merit but does not fully meet PLOS ONE’s publication criteria as it currently stands. Therefore, we invite you to submit a revised version of the manuscript that addresses the points raised during the review process.

We look forward to receiving your revised manuscript.

Kind regards,

Ditta Zobor, MD

Academic Editor

PLOS ONE

4. We note that Figure 4 and 10 in your submission contain copyrighted images. All PLOS content is published under the Creative Commons Attribution License (CC BY 4.0), which means that the manuscript, images, and Supporting Information files will be freely available online, and any third party is permitted to access, download, copy, distribute, and use these materials in any way, even commercially, with proper attribution. For more information, see our copyright guidelines: http://journals.plos.org/plosone/s/licenses-and-copyright.

a. You may seek permission from the original copyright holder of Figure 4 and 10 to publish the content specifically under the CC BY 4.0 license. 

5. Please include a copy of Table 2 which you refer to in your text on page 10.

Reviewers' comments:

Reviewer's Responses to Questions

**Comments to the Author**

1. Is the manuscript technically sound, and do the data support the conclusions?

Reviewer #1: Yes

Reviewer #2: Yes

2. Has the statistical analysis been performed appropriately and rigorously? 

Reviewer #1: Yes

Reviewer #2: Yes

3. Have the authors made all data underlying the findings in their manuscript fully available?

Reviewer #1: Yes

Reviewer #2: No

4. Is the manuscript presented in an intelligible fashion and written in standard English?

Reviewer #1: Yes

Reviewer #2: Yes

5. Review Comments to the Author

Reviewer #1: The authors created a setup for evaluating the detection threshold of spatio-temporal artefacts in movies.

and developed a mathematical model to predict those threshold. Furthermore, they conducted a comprehensive

study to collect data, which in turn was used to test the model. Finally, the results were compared to existing

models.

This is really an impressive piece of work and I would like to congratulate the authors to it!

Accordingly, I only have some minor comments:

line 73: reference Cecchi [2018] is missing

line 118 - 155: These paragraph my better fit into the discussion section

line 185: Why were especially these brightness values chosen?

line 335: Table reference missing (??)

Reviewer #2: The summary of the research and overall impression is good, however, there is a lot of scope for improvement in the article for a better understanding of the reader. The strength of the article is the success in creating the model as expected, somewhere the methodology has been made complex by connecting it to the introduction part. This part has to be reworked majorly in the current article.

The authors have made a good start in the abstract and introduction part of the article, however, in comparing the research question to the previously published article, the results of the current study have also been discussed.

Figure 1 to 5 requires citation and repositioning in the text of the article.

Line 67, figure 6 needs to be repositioned. Line 75-80 require a reframing of the sentence for a better understanding of the reader. Line 117, the same word has been repeated?

While the study appears to be good, I advise the authors to rework the methodology part of the study, giving better clarity for the current work being published.

Needs clarification on what basis the sample size for both the 1st and 2nd experiments was decided.

The methodology followed for the comparison of the model needs an explanation of the procedure/methods.

Values excluded from the study are missing in the result part, although has been discussed in the discussion part with justification.

Points in lines 295-297 can be a part of the discussion.

Line 363, is that Table 1? In the content, it has been written as table 2, but there is no table 2 found in the present article.

Table 1, representation of data in comparison to other models, PLCC need to be written as Pearson Linear Correlation Coefficient

Discussion can be elaborated, by moving some of the points from the previous parts of the article. Authors should have a comparison of the current model with the previous ones in the discussion part as this is not one of the main objectives as per the title.

Limitations of the study can be discussed.

Full data sheet access has not been attached to the article.

The reference style has to be reworked as there is a discrepancy in the list of references.

6. PLOS authors have the option to publish the peer review history of their article (what does this mean?). If published, this will include your full peer review and any attached files.

Reviewer #1: No

Reviewer #2: No

---

## [Author Response · Author response to Decision Letter 0]

9 Apr 2024

We thank the reviewers for their positive appraisals of our submission and the editor for the opportunity to revise our work.

---

## [Decision Letter · Decision Letter 1]

6 May 2024

A contrast sensitivity model of the human visual system in modern conditions for presenting video content.

PONE-D-23-30415R1

Dear Dr. Mozhaeva,

We’re pleased to inform you that your manuscript has been judged scientifically suitable for publication and will be formally accepted for publication once it meets all outstanding technical requirements.

Kind regards,

Ditta Zobor, MD

Academic Editor

PLOS ONE

Reviewers' comments:

Reviewer's Responses to Questions

**Comments to the Author**

1. If the authors have adequately addressed your comments raised in a previous round of review and you feel that this manuscript is now acceptable for publication, you may indicate that here to bypass the “Comments to the Author” section, enter your conflict of interest statement in the “Confidential to Editor” section, and submit your "Accept" recommendation.

Reviewer #2: All comments have been addressed

2. Is the manuscript technically sound, and do the data support the conclusions?

Reviewer #2: Yes

3. Has the statistical analysis been performed appropriately and rigorously? 

Reviewer #2: Yes

4. Have the authors made all data underlying the findings in their manuscript fully available?

Reviewer #2: Yes

5. Is the manuscript presented in an intelligible fashion and written in standard English?

Reviewer #2: Yes

6. Review Comments to the Author

Reviewer #2: The suggested modifications have proven to be highly effective. I commend the authors for their efforts in enhancing the article to make it more accessible to readers. It might be beneficial to consider relocating Figure 11 to the discussion section for added clarity and relevance.

7. PLOS authors have the option to publish the peer review history of their article (what does this mean?). If published, this will include your full peer review and any attached files.

Reviewer #2: **Yes: **NAMRATHA

---

## [Editor Report · Acceptance letter]

9 May 2024

PONE-D-23-30415R1 

PLOS ONE

Dear Dr. Mozhaeva, 

I'm pleased to inform you that your manuscript has been deemed suitable for publication in PLOS ONE. Congratulations! Your manuscript is now being handed over to our production team.

Kind regards, 

on behalf of

Dr. Ditta Zobor 

Academic Editor

PLOS ONE